# Comparison of Estetrol Exposure between Women and Mice to Model Preclinical Experiments and Anticipate Human Treatment

**DOI:** 10.3390/ijms24119718

**Published:** 2023-06-03

**Authors:** Anne Gallez, Gwenaël Nys, Vincent Wuidar, Isabelle Dias Da Silva, Mélanie Taziaux, Virginie Kinet, Ekaterine Tskitishvili, Agnès Noel, Jean-Michel Foidart, Géraldine Piel, Marianne Fillet, Christel Péqueux

**Affiliations:** 1Laboratory of Biology, Tumor and Development Biology, GIGA-Cancer, University of Liège, B23, Avenue Hippocrate 13, 4000 Liège, Belgium; 2Center for Interdisciplinary Research on Medicines (CIRM), Laboratory for the Analysis of Medicines, University of Liège, Avenue Hippocrate 15, 4000 Liège, Belgium; 3Mithra Pharmaceuticals, Rue Saint-Georges 5/7, 4000 Liège, Belgium; 4Center for Interdisciplinary Research on Medicines (CIRM), Laboratory of Pharmaceutical Technology and Biopharmacy, University of Liège, Avenue Hippocrate 15, 4000 Liège, Belgium

**Keywords:** estetrol, exposure, pharmacokinetics, human, mice, route of administration

## Abstract

Estetrol (E4) is a natural estrogen with promising therapeutic applications in humans. The European Medicines Agency and the Food and Drug Administration have approved the use of 15 mg E4/3 mg drospirenone for contraceptive indication. Phase III clinical trials with 15–20 mg E4 for the relief of climacteric complaints are currently running. Relevant data from preclinical animal models are needed to characterize the molecular mechanisms and the pharmacological effects of E4 and possibly to reveal new therapeutic applications and to anticipate potential adverse effects. Therefore, it is important to design experimental procedures in rodents that closely mimic or anticipate human E4 exposure. In this study, we compared the effects of E4 exposure after acute or chronic administration in women and mice. Women who received chronic E4 treatment per os at a dose of 15 mg once daily reached a steady state within 6 to 8 days, with a mean plasma concentration of 3.20 ng/mL. Importantly, with subcutaneous, intraperitoneal or oral administration of E4 in mice, a stable concentration over time that would mimic human pharmacokinetics could not be achieved. The use of osmotic minipumps continuously releasing E4 for several weeks provided an exposure profile mimicking chronic oral administration in women. Measurements of the circulating concentration of E4 in mice revealed that the mouse equivalent dose necessary to mimic human treatment does not fit with the allometric prediction. In conclusion, this study highlights the importance of precise definition of the most appropriate dose and route of administration to utilize when developing predictive preclinical animal models to mimic or anticipate specific human treatment.

## 1. Introduction

Estetrol (E4), a natural estrogen with four hydroxyl groups, was discovered in 1965 by Diczfalusy [1]. This natural hormone is produced by the human fetal liver (both male and female) during pregnancy and is no longer detectable soon after birth. In all other species tested so far (i.e., rats and mice), levels of E4 were undetectable [2]. E4 is an end product of steroid metabolism, meaning that there is no metabolism backwards to estriol (E3), estradiol (E2) or estrone (E1) [3]. Nevertheless, its physiological role during pregnancy remains unknown.

Exogenous E4 administration in rodents revealed that it has several estrogen-like effects on numerous tissues in common with E2 and E3. E4 exerts favorable effects on the central nervous system and was able to prevent the appearance of vasomotor symptoms in an animal model [4]. The neuroprotective effects displayed by E4 are related to its capacity to induce the synthesis of allopregnanolone [5] and β-endorphin [6]. Furthermore, E4 has been shown to attenuate brain injury in a rat neonatal model of hypoxic–ischemic encephalopathy [7,8,9]. Like E2, it displays an estrogenic effect on the vaginal epithelium [10], prevents atheroma [11,12] and modulates endothelial functions in mice [11,13,14,15]. E4 prevents osteoporosis by increasing bone mineral density in female rats and by stimulating the proliferation of human osteoblasts [16,17]. Moreover, it is a weak estrogen on mouse mammary glands [18] and has a neutral effect on breast cancer growth at a potential therapeutic dose for menopause, even when combined with progesterone or drospirenone (DRSP) [19,20].

Preclinical experiments have been conducted using distinct routes of administration such as oral gavage, intraperitoneal injection, subcutaneous pellet or osmotic pump or via addition of the drug to food. The route of administration of a steroid such as E2 largely influences its pharmacokinetics (PK) [21,22]; therefore, it can be anticipated that the PK of E4 will also vary with this parameter. Since mice are an excellent model for studying the effects of drugs or treatments, it is crucial to properly define E4 PK in mice. In particular, patient-derived xenografts (PDX) have emerged as a promising tool allowing for the treatment of human tissue samples (cancers or healthy tissues) in in vivo conditions [23]. PDX models involve transplanting human tissue directly into immunodeficient mice, submitting human tissue to blood circulating levels of the tested drug. Overall, PDX models offer a clinically relevant approach to preclinical drug testing, which may lead to improved clinical trial design and more effective treatments.

Thus, a better understanding of E4 PK in women and mice can allow for the appropriate translation of preclinical data obtained from these models to humans and enable the adequate design of preclinical mouse experiments in which it is essential to administer E4 in a pattern that closely mimics or anticipates E4 exposure in women. For example, such experiments are particularly important to anticipate potential harmful effects that, without animal models, can only be identified during patient follow-up over decades.

On the other hand, E4 has characteristics distinct from those of other estrogens. E4 has the longest half-life of naturally occurring estrogens: up to 28 h (in contrast, for example, to E3: t_1/2_ = 10–20 min) after oral administration in humans. This property, together with its high bioavailability of more than 70%, makes E4 suitable as an oral drug, especially for once-daily oral administration [24,25]. More importantly, combined with DRSP, E4 showed reduced hemostatic effects as compared with ethinylestradiol (EE)/DRSP combinations in women in clinical trials [26,27]. E4 does not stimulate the synthesis of coagulation factors by the liver, nor does it mitigate natural anticoagulants, suggesting a limited impact on venous thromboembolic risk for women taking E4 compared with women taking E2- and EE-containing combined oral contraception (COC) [28,29]. The European Medicines Agency and the Food and Drug Administration have approved the use of E4 combined with DRSP as a new generation of COC [26,27,30,31,32,33,34,35,36].

Interestingly, the characteristics of E4 make it a potentially appropriate compound to be used for additional indications such as menopause hormone therapy (MHT) [25] or prevention of hypoxic–ischemic encephalopathy in newborns [7,8,9]. The use of E4 for these indications requires different formulations or routes of administration depending on the specificity of the treatment and its tissue target. The optimal use of hormone therapy requires an extensive understanding of pharmacology. Therefore, to properly mimic or anticipate potential new treatments in humans, the optimal use of E4 requires a high level of pharmacological knowledge in mice.

To decipher the molecular mechanisms and pharmacological effects of E4 and to reveal new therapeutic applications or adverse effects, further robust preclinical studies in animals, such as mice, are still needed. Because of the intrinsically faster clearance in rodents, the half-life of E4 differs greatly between humans (28 h) and rats (2–3 h) [17,24,37]. In an attempt to mimic or anticipate human treatments, it is important to better characterize and compare the different routes of administration that could be used to study E4 in mice. In this work, we compared the exposure profiles obtained in women under acute or chronic oral treatment with E4 to those obtained in mice when E4 was administered intravenously (i.v.), subcutaneously (s.c.), intraperitoneally (i.p.), by oral gavage or continuously with Alzet^®^ osmotic minipumps.

## 2. Results

### 2.1. E4 PK in Humans

After a single administration of E4 at a dose of 5, 15 or 45 mg in tablet form, plasma E4 C_max_ values were 6.71 ± 3.47 ng/mL, 20.12 ± 9.73 ng/mL and 56.59 ± 14.51 ng/mL (mean ± SD, *n* = 9), respectively. C_max_ was achieved 19 to 30 min after administration (t_max_, Table 1). This absorption phase is generally followed by a decline, then by an increase due to secondary reabsorption. The monoexponential apparent terminal phase associated with E4 elimination emerged at approximately 24 h post dose, when the earlier processes of E4 absorption, distribution throughout the body and reabsorption were complete (Figure 1A). The AUC0-∞ values were 38.27 h·ng/mL, 90.09 h·ng/mL and 321.05 h·ng/mL, respectively for 5, 15 and 45 mg E4 single doses. The apparent terminal half-life (t_1/2_) values were 28.21 h, 19.22 h and 26.64 h, respectively, corresponding to a mean apparent terminal half-life (t_1/2_) of 24.69 h.

With multiple once-daily administrations of 15 mg E4 tablets for 14 days, a steady state was reached within 6 to 8 days (Figure 1B). The average concentration (C_av_) was 3.20 ± 0.92 ng/mL (mean ± SD, *n* = 19, Table 2). The AUC0-τ (τ = 24 h) was 76.79 h·ng/mL. There was no increase in the peak E4 systemic exposure, and a limited (less than twofold on average) increase in the extent of E4 exposure compared to the single administration of the same dose.

### 2.2. E4 PK after Acute Administration in Mice

Intravenous (i.v.) injection of E4 (7.5 µg, 0.3 mg/kg) in the tail vein led to a high circulating concentration of E4 followed by a rapid decrease (Figure 2), reaching the detection limit of the assay 6 h after administration. The PK parameters are summarized in Table 3. The extrapolated blood concentration calculated for t = 0 (C0) was 294.40 ng/mL, and the half-life (t_1/2_) was 120 min.

Subcutaneous (s.c.) and intraperitoneal (i.p.) injections of E4 (7.5 µg, 0.3 mg/kg) showed a sharp peak delivery profile (Figure 3A,B). The C_max_ was reached within 10 min, with values of 90.92 ± 20.83 ng/mL (mean ± SD, *n* = 15) and 70.54 ± 9.90 ng/mL (mean ± SD, *n* = 6) for s.c. and i.p. administration, respectively (Table 1). The AUC0-∞ values of these two routes of administration were close (97.42 and 68.17 h·ng/mL, respectively) and in the same range as those obtained for i.v. injection (75.57 h·ng/mL), corresponding to 100% bioavailability.

In contrast, oral gavage induced a biphasic profile with two peaks appearing after 15 min and 60 min (Figure 3C). C_max_ reached 11.5 ± 5.3 ng/mL (mean ± SD, *n* = 6), and the AUC0-∞ was approximately three times lower than those obtained with i.v., s.c. or i.p. administrations (Table 1 and Table 3), corresponding to approximatively 33% bioavailability.

### 2.3. E4 PK with Chronic Delivery in Mice

E4 was continuously administered with Alzet^®^ osmotic minipumps delivering 3 doses of E4: 0.1, 0.3 or 1 mg/kg/day. E4 blood concentration was measured 24 h and 1, 3 and 5 weeks after treatment initiation (Figure 4A). One day after administration, blood concentrations of E4 were 2.74 ± 1.23 ng/mL (mean ± SD, *n* = 6), 8.33 ± 3.50 ng/mL (mean ± SD, *n* = 6) and 22.68 ± 2.27 ng/mL (mean ± SD, *n* = 6) for 0.1, 0.3 and 1 mg/kg/day doses, respectively. This route of administration provided a constant level of circulating E4 across the entire duration of the treatment (Figure 4B). E4 blood concentrations detected after 1, 3 and 5 weeks of treatment are reported in Table 4. The overall mean blood concentrations of E4 during the treatment were 2.47 ± 0.88 ng/mL (mean ± SD, *n* = 24), 6.49 ± 2.75 ng/mL (mean ± SD, *n* = 24) and 26.14 ± 8.49 ng/mL (mean ± SD, *n* = 24) for doses of 0.1, 0.3 and 1 mg/kg/day, respectively (Table 2). E4 mean blood circulating concentrations were proportional to the administered E4 dose, r^2^ = 0.997 (Figure 4C). The AUC0-τ values were determined for a τ period of 24 h and reached 58.56, 143.57 and 693.60 h·ng/mL for E4 0.1, 0.3 and 1 mg/kg/day, respectively. Using the same procedure, the potential impact of the vehicle on E4 PK profiles was evaluated by comparing dilution of E4 (0.3 mg/kg/day) in HP-β-CD or in PPG (Figure 4D). There was no statistical difference between the PK measurements obtained with either preparation. After 24 h of exposition (T_max_), a constant concentration was reached: 5.01 ± 4.44 ng/mL (mean ± SD, *n* = 6) and 8.33 ± 3.50 ng/mL (mean ± SD, *n* = 6) when E4 was diluted in HP-β-CD or in PPG, respectively. The pharmacokinetic parameters (Table 5) show comparable AUC0-τ values determined for a τ period of 24 h for both formulations. The C_av_ values were 6.66 ± 2.43 ng/mL (mean ± SD, *n* = 25) and 6.22 ± 3.16 ng/mL (mean ± SD, *n* = 27) when E4 was diluted in HP-β-CD or in PPG, respectively.

## 3. Discussion

E4 is a promising native estrogen for clinical indications [25]. In this study, we evaluated the exposure of E4 in women through single or multiple once-daily oral dose administration and compared them to E4 exposure obtained in mice following several routes of administration. The half-life measured in women and mice was 28 h and 2 h, respectively, which is same range as previously reported and following the allometric correlation [17,24,37].

C_max_ and AUC0-∞ values obtained for a single-dose administration of E4 to women were proportional to E4 doses of 5, 15 or 45 mg. In contrast to mice, the circulating concentration of E4 was still detectable after 24 h in women. The monoexponential apparent terminal phase associated with E4 elimination emerged only approximately 24 h post dose. When women received a multiple once-daily oral administrations of 15 mg E4, a steady state was reached within 6 to 8 days of treatment. This is in agreement with a previous PK study performed in postmenopausal women who were administered multiple increasing doses of 10, 20 or 40 mg E4 per os [37].

The route of administration of a steroid largely influences its PK, pharmacodynamic, bioavailability, hormonal activity and metabolism [38]. In mice, the AUC0-∞ values obtained from i.v., s.c. and i.p. injections of 7.5 µg E4 (0.3 mg/kg) were in the same range and provided 100% bioavailability. When E4 was administered by s.c. or i.p. injections, the PK profiles were similar and showed a sharp peak with comparable C_max_ values (70–90 ng/mL). C_max_ was reached after 10 min; then, the circulating E4 concentration decreased rapidly to reach 30% of C_max_ after 45 min. The circulating levels of E4 then decreased more slowly to reach the detection limit of the assay after 6 h. Based on these parameters and in contrast to what we observed in women, it is not possible to reach a stable circulating concentration in mice, even with multiple once-daily s.c. or i.p. administrations. These results indicate that by encompassing the enterohepatic circulation, s.c. and i.p. E4 injections provide high and sharp C_max_ values, followed by a rapid decrease. In mice, s.c. and i.p. injections appear to be the most appropriate routes to induce a bolus-like or pulsed exposure that mimics intranasal or sublingual treatments in women [38].

When E4 was administered to mice by oral gavage, the PK profile was distinct from that obtained with s.c. or i.p. injections. We observed a biphasic profile despite the absence of enterohepatic recirculation in mice. The AUC0-∞ obtained with the oral gavage was 3-times lower than with s.c. or i.p. injections, corresponding to approximatively 33% relative bioavailability. C_max_ was seven times lower than with s.c. or i.p. injections for the same administered dose. After 2 h, the circulating E4 concentration decreased by 83% and became undetectable after 6 h. These results highlight that in mice, administration of E4 by oral gavage does not allow a stable circulating concentration to be reached. This is in contrast with what we observed in humans.

The PK profiles observed in mice for s.c. or oral gavage administration were rather similar to the PK profiles described in rats by Coelingh Bennink et al. [17]. The half-life in rats is 2–3 h and 2 h in mice. The C_max_ values measured in rats for a dose of 0.5 mg/kg administered by s.c. injection or by oral gavage were 86.5 ng/mL and 52 ng/mL, respectively. We measured C_max_ values of 90.92 ± 20.83 ng/mL and 11.52 ± 5.25 mg/mL in mice for a dose of 0.3 mg/kg administered by s.c. injection or by oral gavage, respectively. Depending on the administered dose, the oral bioavailability in rats was estimated to be 37–70% of s.c.; in mice, we observed an oral bioavailability of 36% of s.c. The t_max_ measured in mice occurred earlier (10 min) than in rats (30 min); however, E4 plasma level sampling in rats started no earlier than 30 min after administration.

The impact of the vehicle in which a drug is administered is often underestimated in the literature. Nevertheless, this point can be critical to understand drug effects. In this study, we compared exposure profiles obtained by Alzet^®^ osmotic minipump administration of E4 diluted either in PPG or in HP-β-CD, two vehicles commonly used for steroid preparations. Exposure profiles and PK parameters were in the same range, without any statistical difference, irrespective of the nature of the vehicle (PPG or HP-β-CD).

In comparison to i.v., s.c. and i.p. injections of the same dose of E4 (0.3 mg/kg) per day, the continuous administration of E4 by Alzet^®^ osmotic minipumps resulted in a lower but more stable circulating E4 concentration of 6.49 ± 2.75 ng/mL (C_av_, mean ± SD) and an AUC0-τ (τ = 24 h) of 143.57 h·ng/mL. The circulating concentration (C_av_) obtained with Alzet^®^ osmotic minipump administration (0.3 mg/kg) was closer to the C_max_ value obtained with oral gavage (11.52 ± 5.25 ng/mL (mean ± SD)) in comparison to s.c. or i.p. administrations. However, with Alzet^®^ osmotic minipumps, the circulating concentration was constant, while oral gavage did not allow a stable concentration to be achieved. In comparison, a chronic administration of E4 (15 mg, 0.25 mg/kg) to women with multiple once-daily oral treatment provided an AUC0-τ (τ = 24 h) of 76.79 h·ng/mL. A dose of 0.3 mg/kg/day E4 administered by Alzet^®^ osmotic minipumps allowed for systemic blood exposure in mice two times higher than that elicited in women with daily E4 administration of 15 mg (0.25 mg/kg).

When comparing these results to the equivalent animal dose calculated by allometric correlation defined by Nair and Jacob [39] (Table 6), we surprisingly observed that the mouse equivalent dose (MED) predicted by this allometric correlation does not correspond to the circulating levels of E4 measured in this study. Allometric prediction suggests that the blood exposure in mice should be 12 times lower than that in humans, although this study shows a systemic blood exposure in mice two times higher than that elicited in women. These discrepancies may have resulted from the use of subcutaneous administration through Alzet^®^ osmotic minipumps to reach a stable circulating concentration in mice, although in women, E4 was administered orally. Complementary experiments that are beyond the scope of this study should be performed to better understand these discrepancies. Nevertheless, these results suggest that in order to mimic a dose of E4 administered in humans in mice, it is necessary to use a dose 10 times lower than that predicted by the allometric correlation of Nair and Jacob [39].

## 4. Materials and Methods

### 4.1. Human Study Design and Ethical Statement

This study (EudraCT number: 2016-001808-32; clinicaltrials.gov: NCT03075956) was conducted between Jan 2017 and August 2017 at MC Comac Medical Ltd., Sofia, Bulgaria, and complied with the last revision of the Declaration of Helsinki and the ICH guidelines for Good Clinical Practice. This study was an open-label, single-center, randomized, two-period study conducted to characterize the PK of E4 after single and multiple oral doses in healthy female volunteers between 18 and 55 years of age. The enrolment was homogeneous regarding menopausal status. At least 11 premenopausal and 11 postmenopausal women were enrolled (for details, see Appendix A). All participants had an intact uterus, except one, who was hysterectomized in 2017. For endometrial safety, progestin therapy (dydrogesterone 10 mg) was administered once daily for 14 days after period 2, day 15. This therapy was started on the day after the last investigational intake of the medicinal product (E4). The inclusion/exclusion criteria are available in Appendix A. The primary objective of this study was to determine the PK profile of E4 after a single oral dose of 5, 15 or 45 mg (period 1) and the PK profile of E4 after multiple oral doses of 15 mg E4 during a 14-day period (period 2). The study was approved by the Independent Ethics Committee at MC Comac Medical Ltd. (Sofia, Bulgaria).

### 4.2. Participants, Duration of Treatment and Blood Sampling

Inclusion and exclusion criteria are available in Appendix A.

During period 1, 34 subjects were screened, and 27 subjects were randomized to one of the 3 different dose groups comprising 9 subjects each. Sixteen subjects who participated in period 1 continued the study in period 2. Four additional subjects were screened and included directly in period 2 as per the investigator’s discretion and as agreed with the sponsor (Estetra SPRL, Liège, Belgium) in order to compensate for the drop-out of subjects occurring between period 1 and period 2 and during period 2.

During period 1, participants received a single oral dose of 5, 15 or 45 mg E4. During period 2, participants received multiple once-daily oral doses of 15 mg E4 from day 1 to day 14. At least a 14-day washout was applied between period 1 and period 2.

Immediately after aliquoting plasma, the cryovials were placed in an upright position in the corresponding cryobox and rapidly stored at −15 °C to −30 °C. The most prolonged time between the date of the first sample taken and the date of the last sample processed for analysis was 165 days.

### 4.3. Chemicals, Reagents and Steroids

Estetrol (E4) powder and tablets were supplied by Mithra Pharmaceuticals (Liège, Belgium). Hydroxypropyl-β-cyclodextrins (HP-β-CD) with a degree of substitution (ds) of 0.64 was kindly donated by Roquette Corporate (Lestrem, France). Adding cyclodextrin increases the solubility of E4. The preparations of cyclodextrin and E4 were diluted in physiologic solution (NaCl 0.9%) and filtered through a 0.22 µm hydrophilic mixed cellulose ester membrane (Millex-GS Syringe Filter Unit, Millipore, Merck KGaA, Darmstadt, Germany) before acute injections in mice. For chronic treatments with Alzet^®^ osmotic minipumps (#2006, Charles River, France), E4 was diluted either in HP-β-CD and NaCl 0.9% or in ethanol (VWR Chemicals, VWR International, Leuven, Belgium) and propylene glycol (PPG, Sigma Aldrich, Merck KgaA, Darmstadt, Germany).

### 4.4. Ethical Statement for Animal Studies

All animal experiments were conducted in accordance with the requirements of the Federation of European Laboratory Animal Science Associations (FELASA) and approved by the local ethical committee of the University of Liège. All animals were maintained within the accredited Mouse Facility and Transgenics GIGA platform of the University of Liège (Belgium) in a controlled and enriched environment under pathogen-free conditions. More specifically, mice were kept on a 12 h light/dark cycle at 22 °C with free access to food and water.

### 4.5. Mouse Blood Sampling and Quantitation of E4

Blood (10 µL) was collected using volumetric absorptive microsampling devices (VAMS, Neoteryx^®^, Maastricht, Netherlands). The mouse tail was slightly slashed with a scalpel blade to generate a drop of blood. VAMS was placed in contact with the drop, and blood was absorbed until complete filling of the VAMS. Each sampling was performed in duplicate at each time point. E4 was quantitated from whole blood after VAMS collection and extraction by a validated UHPLC-MS/MS method as previously described [40].

### 4.6. Animal Experiments

Female FVB/N mice bred in the animal facility of the GIGA (University of Liège, Liège, Belgium) were bilaterally ovariectomized at 4 weeks of age under isoflurane anesthesia. The average mouse weight was 25 g. Two weeks after surgery, mice were treated following different routes of administration: intravenous (i.v.; tail vein), subcutaneous (s.c.; in the flank), intraperitoneal injections (i.p.), oral gavage or through Alzet^®^ osmotic minipumps (#2006, Charles River, France). For all treatments, 100 μL of a solution containing 75 μg/mL (0.3 mg/kg) of E4 was injected. Each experimental group comprised 6 mice. Blood was collected in duplicate from 6 mice after 1, 5, 10, 15, 30 and 45 min and 1, 2, 3, 6 and 24 h later. Delivery of E4 (0.1, 0.3 or 1 mg/kg/day) with Alzet^®^ osmotic minipumps inserted subcutaneously on the back of the mouse allowed for a continuous release of E4 for 5 weeks. Blood samples were collected in duplicate from 3 mice 24 h and 1, 3 and 5 weeks after pump implantation.

### 4.7. Statistical Analysis

Statistics of the human studies were generated using Statistical Analysis Software (SAS)^®^ version 9.4 (2013). The descriptive statistics presented herein are area under the curve (AUC0-∞), t_max_, arithmetic mean, standard deviation (SD), median, maximum, geometric mean (GM) and CV% of GM calculated for C_max_ for the single-dose regimen. AUC0-τ (τ = 24 h) and C_av_ are presented for multiple-dose regimens.

For mouse experiments, statistical analyses were carried out with GraphPad Prism 7.0 software (2016). The half-life (t_1/2_), AUC0-∞,volume of distribution (Vd), constant of distribution (Kd), constant of elimination (Ke), clearance (Cl) and concentration for t = 0 (C0) were the parameters studied for i.v. administration. The AUC0-∞, the time to reach maximal concentration (t_max_) and the maximal concentration (C_max_) were determined for s.c., i.p. and administration by oral gavage. The AUC0-τ (τ = 24 h) and the average circulating concentration of E4 reached during the chronic treatment (C_av_) were determined for administration with Alzet^®^ osmotic minipumps.

## 5. Conclusions

Preclinical experiments on rodents were conducted to characterize the E4 molecular mechanisms of action and to reveal new therapeutic applications or potential adverse effects. In order to model human treatments, it is essential to administer E4 to mice in a pattern that closely mimics or anticipates E4 exposure in women and to evaluate its effects on physiology or pathophysiology. The comparative study we performed revealed that the steady state observed in women who received a once-a-day oral treatment was not achievable in mice with chronic once-a-day oral gavage, s.c. or i.p. treatment. The sharp peak induced by s.c. or i.p. injections reflects a bolus or pulsed treatment. Oral administration of E4 in mice more closely reflects a discontinuous treatment, but it cannot be considered a pulsed treatment, since there was no induction of a sharp peak in circulating hormone levels. Therefore, based on our results, it appears that the best way to mimic the steady state observed in women on a once-a-day oral treatment is to administer E4 with osmotic minipumps. Importantly, this study highlights the importance of carefully considering the dose and the route of E4 administration used to design preclinical mouse experiments to properly translate the data to human treatment.

## Figures and Tables

**Figure 1 ijms-24-09718-f001:**
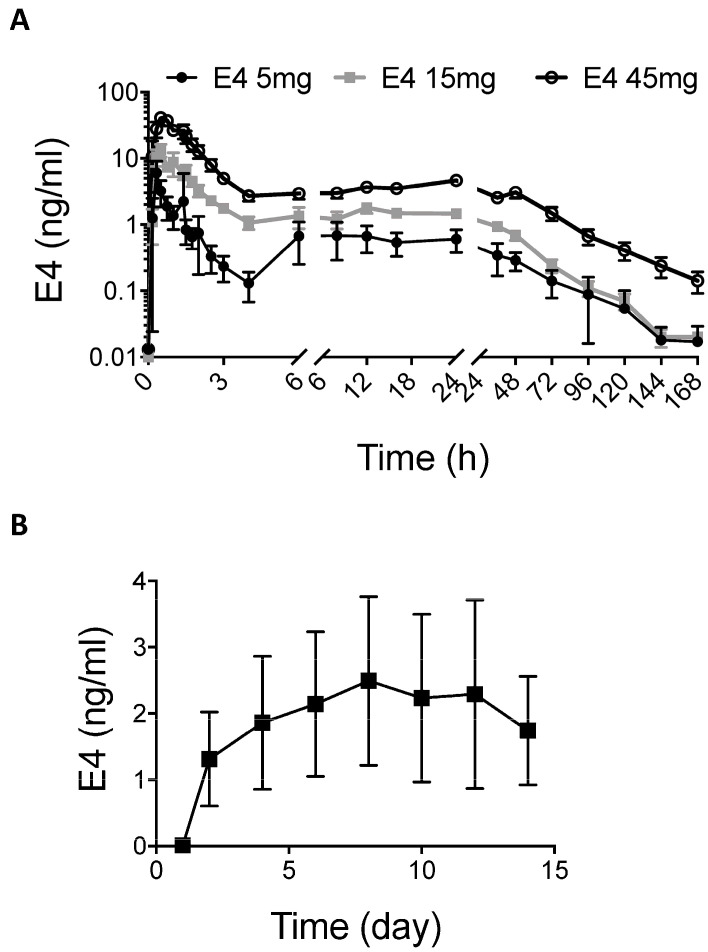
Pharmacokinetics of single- and multiple-dose daily oral E4 treatment in women. (**A**) E4 blood concentration (logarithmic scale, mean ± SD) after a single oral administration of E4 (5, 15 or 45 mg) over time (hours). E4 concentrations (*n* = 9) were measured after 10, 20, 30, 45, 60, 75, 90, 105, 120 and 150 min and after 3, 4, 6, 8, 12, 16, 24, 36, 48, 72, 96, 120, 144 and 168 h. (**B**) E4 blood concentration (mean ± SD, ng/mL) after treatment with multiple daily oral doses of E4 (15 mg) over time (days). E4 concentrations were measured on days 1, 2, 4, 6, 8, 12 and 14 (*n* = 19).

**Figure 2 ijms-24-09718-f002:**
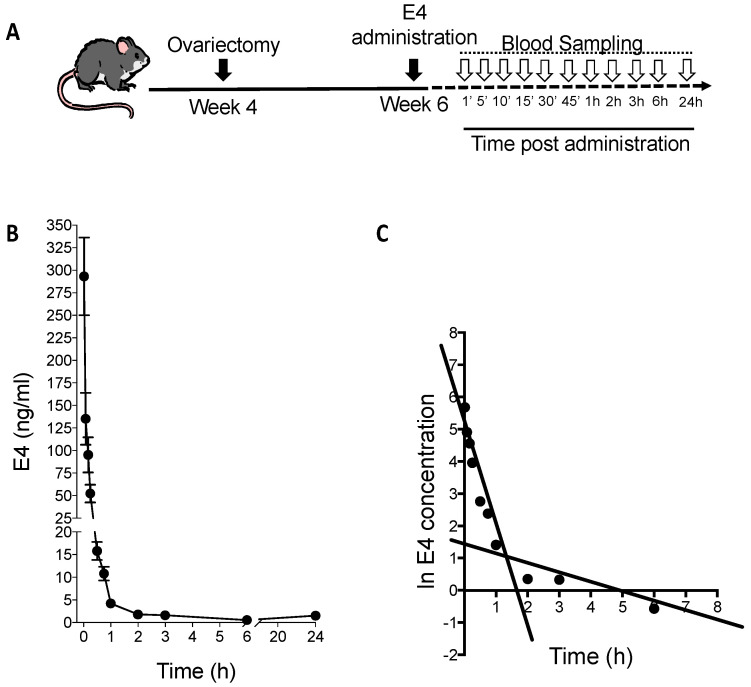
Intravenous administration in mice. (**A**) Schematic treatment protocol of mice: ovariectomy at 4 weeks of age, E4 treatment at 6 weeks of age and blood sampling schedule after treatment. (**B**) Circulating E4 concentrations in blood over time (0 to 24 h) after intravenous injection of 7.5 µg (0.3 mg/kg) E4. Results are expressed in ng/mL as mean ± SD (*n* = 6). (**C**) Expression as a natural logarithm (Ln) of E4 concentrations (ng/mL) in blood over time (h), allowing for definition of the distribution and the elimination phases.

**Figure 3 ijms-24-09718-f003:**
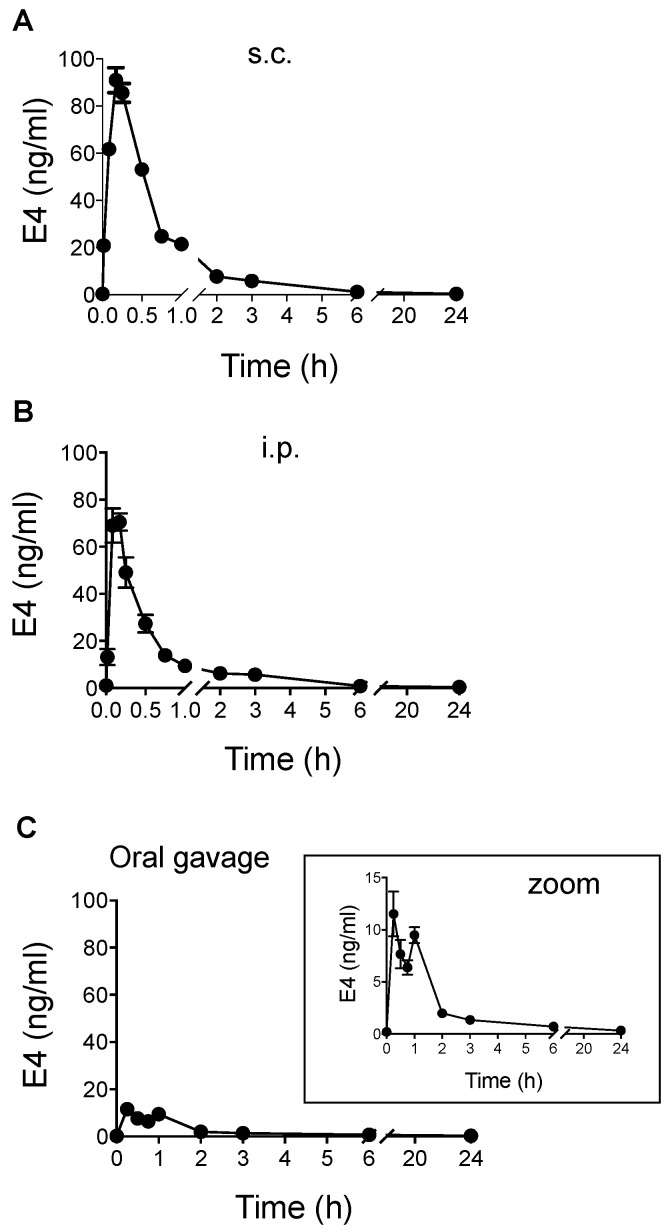
Pharmacokinetics of E4 in mice. E4 blood concentrations (0 to 24 h) after administration of 7.5 µg (0.3 mg/kg) through subcutaneous (s.c., (**A**)), intraperitoneal (i.p., (**B**)) or oral gavage (**C**) routes of administration over time (h). Results are expressed in ng/mL as mean ± SD (*n* = 6).

**Figure 4 ijms-24-09718-f004:**
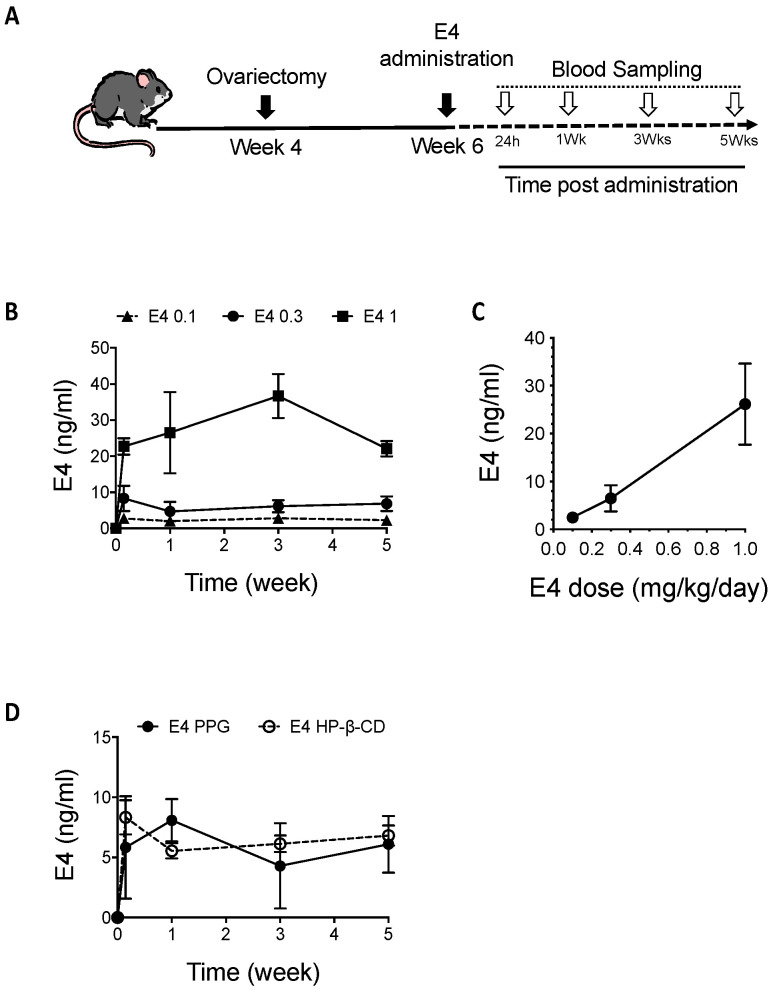
E4 blood plasma levels after Alzet^®^ osmotic minipump administration in mice. (**A**) Schematic treatment protocol in mice: ovariectomy at 4 weeks of age, starting of treatment at 6 weeks of age and blood sampling schedule after starting the treatment. (**B**) Circulating E4 concentrations in blood over time (week): after 24 h, 1 week, 3 weeks and 5 weeks of treatment delivered by Alzet^®^ osmotic minipumps (E4 diluted in PPG). Results are expressed in ng/mL as mean ± SD (*n* = 6). (**C**) Correlation between administered E4 dose and E4 blood circulating concentration. (**D**) Circulating E4 concentrations in blood over time (week): after 24 h, 1 week, 3 weeks and 5 weeks of treatment administered by Alzet^®^ osmotic minipumps (E4 = 0.3 mg/kg/day diluted in HP-β-CD or in PPG). Results are expressed in ng/mL as mean ± SD (*n* = 6).

**Table 1 ijms-24-09718-t001:** Pharmacokinetic parameters of E4 for acute administration, single dose.

Species	Dose	Administration Route	AUC_0-∞_ (h·ng/mL)	t_max_ (min.)	C_max_ (ng/mL)					
					Mean	SD	Median	Max	Geometric Mean	Geometric CV (%)
Human	E4 5 mg	oral (*n* = 9)	38.27	20	6.71	3.47	5.46	12.05	5.92	58.69
	E4 15 mg	oral (*n* = 9)	90.09	30	20.12	9.73	17.70	35.68	17.89	57.56
	E4 45 mg	oral (*n* = 9)	321.05	30	56.59	14.51	56.00	82.38	54.99	25.69
Mice	E4 7.5 µg	s.c. (*n* = 15)	97.42	10	90.92	20.83	85.05	119.70	88.64	1.43
		i.p. (*n* = 6)	68.17	10	70.54	9.90	70.55	85.26	70.08	1.61
		oral gavage (*n* = 6)	30.20	15+60	11.52	5.25	9.00	18.60	10.63	14.39

**Table 2 ijms-24-09718-t002:** Pharmacokinetic parameters of E4 for chronic administration.

Species	Administration Route	AUC_0-τ_ (h·ng/mL)	C_av_ (ng/mL)					
			Mean	SD	Median	Max	Geometric Mean	Geometric CV (%)
Human	Oral, multiple dose E4 15 mg (*n* = 19)	76.79	3.20	0.92	3.06	4.92	3.08	29.76
Mice	Alzet pump E4 0.1 mg/kg/day (*n* = 24)	58.56	2.47	0.88	2.29	5.06	2.32	62.97
	Alzet pump E4 0.3 mg/kg/day (*n* = 24)	143.57	6.49	2.75	6.49	13.23	5.72	32.05
	Alzet pump E4 1 mg/kg/day (*n* = 24)	693.60	26.14	8.49	22.94	42.65	24.88	5.53

**Table 3 ijms-24-09718-t003:** Pharmacokinetic parameters of E4 (i.v.).

Vd	1.80 L
Kd	0.12 min^−1^
C0	294.40 ng/mL
Ke	0.0058 min^−1^
t_1/2_	119.43 min
AUC0-∞	75.57 h·ng/mL
clearance	10.44 mL/min

**Table 4 ijms-24-09718-t004:** E4 blood circulating concentrations under chronic administration by Alzet pump.

	E4 (ng/mL)					
Treatment Doses	E4 (0.1 mg/kg/day) *n* = 6		E4 (0.3 mg/kg/day) *n* = 6		E4 (1 mg/kg/day) *n* = 6	
Treatment Duration (Week)	Mean	SD	Mean	SD	Mean	SD
1	2.04	0.98	4.68	2.69	26.53	11.24
3	2.83	0.49	6.13	1.69	36.66	6.09
5	2.28	0.61	6.82	2.07	22.07	2.14

**Table 5 ijms-24-09718-t005:** Pharmacokinetic parameters of E4 diluted in HP-β-CD or in PPG for chronic administration.

Species	Administration Route	AUC_0-τ_ (h·ng/mL)	C_av_ (ng/mL)					
			Mean	SD	Median	Max	GEOMETRIC MEAN	Geometric CV (%)
Mice	Alzet pump E4 0.3 mg/kg/day in HP-β-CD (*n* = 25)	149.47	6.66	2.43	6.33	13.23	6.25	23.24
	Alzet pump E4 0.3 mg/kg/day in PPG (*n* = 27)	139.73	6.22	3.16	6.50	11.43	4.96	49.25

**Table 6 ijms-24-09718-t006:** MED calculation by allometric correlation.

Human Daily Dose (mg)	Human Dose (mg/kg)	MED (mg/kg)
5	0.08	1
15	0.25	3
45	0.75	9

MED: mouse equivalent dose.

## Data Availability

Data from the PK study conducted in mice will be shared upon request, but data used for the PK study in women remain confidential.

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
