# Peer review of "Comparison of Estetrol Exposure between Women and Mice to Model Preclinical Experiments and Anticipate Human Treatment"

_ijms, 2023, doi:10.3390/ijms24119718_

Round 1

Reviewer 1 Report (New Reviewer)

In the present work, Gallez et al. try to explain Estetrol exposure between women and mice. There are some questions that should be explained.

1. Title should be simplified.

2. The European Medicines Agency and the Food and Drug Administration approved the use 17 of 15mg E4/3mg Drospirenone for contraceptive indication. Why is phase III clinical trials with 15-20 mg E4?

3. Introduction is too long, should be simplified. However, some related references are not included.

Lee A, Syed YY. Estetrol/Drospirenone: A Review in Oral Contraception. Drugs. 2022;82(10):1117-1125.

Kaunitz AM, Achilles SL, Zatik J, Weyers S, Piltonen T, Suturina L, Apolikhina I, Bouchard C, Chen MJ, Jensen JT, Westhoff CL, Jost M, Foidart JM, Creinin MD. Pooled analysis of two phase 3 trials evaluating the effects of a novel combined oral contraceptive containing estetrol/drospirenone on bleeding patterns in healthy women. Contraception. 2022;116:29-36.

Klipping C, Duijkers I, Mawet M, Maillard C, Bastidas A, Jost M, Foidart JM. Endocrine and metabolic effects of an oral contraceptive containing estetrol and drospirenone. Contraception. 2021;103(4):213-221.

4. Lines 128-129, ‘The study (EudraCT number: 2016-001808-32; clinicaltrials.gov: NCT03075956) was conducted between Jan 2017 and August 2017 at the MC Comac Medical Ltd, Sofia,’. It is five years ago.

5. Line 133, ‘healthy female volunteers between 18 and 55 years of age’. Age span is too large.

6. Format of tables should be revised.

7. Discussion section, there are only three references that are related to this study.

8. Format of most references should be revised. For example, References 6, 12-14, 16, 18, 20, 23, 24, 28, 33, 34.

Moderate editing of English language.

Author Response

Reviewer 2 Report (New Reviewer)

This manuscript reported comparison of estetrol exposure between women and mice to model preclinical experiments and anticipate human treatment which is fit for IJMS journal. No concern for this manuscript.

Author Response

We thank the reviewer for the revision of our manuscript.

Reviewer 3 Report (New Reviewer)

Comments about the manuscript:

“Comparison of Estetrol exposure between women and mice to model preclinical experiments and anticipate human treatment”

Estetrol (E4) is an estrogen that can be used as a contraceptive. Clinical trials for the relief of some disorders are currently underway. Tests on animal models are necessary before clinical trials, which requires the development of experimental procedures in these models, in this case rodents. In this manuscript, the authors compared the effects obtained after acute or chronic administration of E4 in women and in mice. The women received chronic treatment with E4 once a day. In mice, only the use of osmotic minipumps releasing E4 continuously for several weeks, made it possible to obtain a result mimicking chronic oral administration in women. Finally, this study highlighted the importance of precisely preparing the dose and the route of administration of the molecule in predictive preclinical animal models.

This study seems important to me and deserves to be published. The manuscript is well written, clear with very many results. I have few minor remarks.

Page 3, lines 134-135. “At least 11 premenopausal and 11 postmenopausal women were enrolled.”: The number of women with their age and their characteristics could be useful: according to the text, we do not know exactly how many women were registered in each category (at least 11 does not seem sufficiently precise to me). Maybe a table would help.

Page 4, line 171. “All animals were maintained”: what were the rearing conditions for the mice? (temperature, nycthemeral rhythm, food...)

Page 8, line 262. Write “(Tables 1 and 3)” instead of “(Table 1 and 3)”

Author Response

Reviewer 4 Report (New Reviewer)

An important study providing critical data on the exposure of model mice and women to estetrol in a preclinical study with the aim of improving its use in human treatment. In general, the manuscript is well-written and easy to follow and I have only made a few suggestions in the edited text to help improve on the quality of the manuscript.

The quality of English Language is good and helps understand what the authors are communicating including the science.

Author Response

We thank the reviewer for the suggestions made throughout the text, especially regarding English editing. To address them, modifications in the manuscript have been highlighted in red.

Round 2

Reviewer 1 Report (New Reviewer)

Thanks for author’s responses. However, the style of Title is not suitable for IJMS, and the format of references should be revised. For example,

Reference 1, ‘Hagen, A. A.; Barr, M.; Diczfalusy, E., Metabolism of 17-Beta-Oestradiol-4-14-C in Early Infancy. Acta endocrinologica 1965, 49, 207-20’ should been revised as: Hagen, A. A.; Barr, M.; Diczfalusy, E., Metabolism of 17-beta-oestradiol-4-14-C in early infancy. Acta Endocrinol, 1965, 49, 207-20.

Reference 7, ‘Tskitishvili, E.; Nisolle, M.; Munaut, C.; Pequeux, C.; Gerard, C.; Noel, A.; Foidart, J. M., Estetrol attenuates neonatal hypoxic-ischemic brain injury. Experimental neurology 2014, 261C, 298-307. ’ should been revised as: Tskitishvili, E.; Nisolle, M.; Munaut, C.; Pequeux, C.; Gerard, C.; Noel, A.; Foidart, J. M., Estetrol attenuates neonatal hypoxic-ischemic brain injury. Exp Neurol, 2014, 261, 298-307.

Please check the references throughout.

Minor editing of English language required

This manuscript is a resubmission of an earlier submission. The following is a list of the peer review reports and author responses from that submission.

Round 1

Reviewer 1 Report

The work is interesting but needs further investigation to relate experiments on mice with possible use in women. For a better understanding of the text it is suggested to write materials and methods before results and discussion. On line 173 it is unclear whether the values refer to doses or weeks.

Author Response

Point by point answer to reviewers

Reviewer #1

The work is interesting but needs further investigation to relate experiments on mice with possible use in women.

The purpose of this study is precisely to relate data obtained from experiments on mice with possible use in women. Indeed, conducting preclinical studies on mice before or in the course of proceeding to clinical studies on humans is crucial for several reasons. Firstly, mice share many genetic and physiological similarities with humans, making them an excellent model for studying the effects of drugs or treatments. Especially, patient-derived xenografts (PDX) have emerged as a promising tool allowing the treatment of human tissue samples (cancers or healthy tissues) in in vivo conditions, predicting drug potency. PDX models involve transplanting human tissue directly into immunodeficient mice, submitting human tissue to blood circulating levels of the tested drug. This maintains the original features of patient’s sample and reflects drug sensitivity. Overall, PDX models offer a more clinically relevant approach to preclinical drug testing, which may lead to improved clinical trial design and more effective treatments. Additionally, preclinical studies allow researchers to identify potential safety concerns, such as toxicity or adverse effects, before testing on human subjects. This approach not only ensures the safety of human participants but also saves time and resources by eliminating drugs or treatments that are ineffective or unsafe early on in the development process. Overall, preclinical studies on mice are essential for establishing the safety and efficacy of potential drugs or treatments before they are tested on human subjects, allowing anticipation of potential adverse effects.

More specifically, mice are widely used in reproductive research because they have many similarities with humans in terms of reproductive anatomy, physiology, and genetics. Mice have a short gestational period, which means that researchers can study multiple generations of mice in a relatively short amount of time. They are also relatively easy to manipulate genetically, which allows researchers to create mouse models of human reproductive disorders and diseases. These models can be used to study the underlying mechanisms of these disorders and to test potential treatments. Additionally, the use of mice in research allows for more controlled experiments than would be possible in human studies, reducing variability and increasing the accuracy of results. Overall, the use of mice in reproductive research has led to significant advancements in our understanding of the female reproductive system and has facilitated the development of new treatments and therapies for reproductive disorders and diseases.

To reveal new therapeutic applications in humans, to anticipate potential adverse effects and to characterize the complex molecular mechanisms and the pharmacological effects of estetrol, the accumulation of relevant data from preclinical animal models and translational research is still mandatory. In this context, our study is particularly relevant since it highlights the importance to carefully consider the dose and the route of E4 administration used to design preclinical mouse experiments. The data we are presenting will allow the proper design of preclinical mouse experiments in which it is essential to administer estetrol to mice in a pattern that closely mimics or anticipate estetrol exposure in women. Therefore, to anticipate potential new treatments or adverse effect in human, the optimal use of hormone therapy requires a high level of pharmacological knowledge in mice as well. This will favor the transposition of results obtained from mice to human.

To clarify this issue, we modified the title of the manuscript and we introduced some precision in the introduction. Modifications in the manuscript are highlighted in red.

For a better understanding of the text it is suggested to write materials and methods before results and discussion.

During the redaction of the manuscript, I used the template provided by IJMS as requested by the instruction to authors. Nevertheless, to meet reviewer comment, the materials and methods section has been moved before the results section.

On line 173 it is unclear whether the values refer to doses or weeks.

In the document provided by the Editor, line 173 refers to: “E4 blood concentration was measured 24h, 1, 3 and 5 weeks after treatment initiation (Fig. 4A). One day after administration, blood concentrations of E4 ». As mentioned in the text, the values refer to weeks.

Reviewer 2 Report

  In this study, Gallez et al. assess E4 exposure in women through single or multiple oral doses once daily and compare them with E4 exposure obtained in mice following various routes of administration. The main finding of their results indicates that to mimic in mice the dose ofE4 administered to humans, it is necessary to use a dose 10 times lower than that predicted by the Nair and Jacob allometric correlation.   The main concern of the work is the relevance of the results and the biological significance of them. In general terms, and from a clinical point of view, the experimental design goes from mice validation to human test through the very well known and stratified phases of human trials. However, in this work the experimental research flow goes from humans to mice without a rationale thought.   In this regard and based on the data generated through experimental animal models and various clinical studies making use of variable doses and routes of administration, the use of E4 combined with DRPS has already been approved by the European Medicines Agency and the Food and Drug Administration as a new combined oral contraception generation. The data provided in this article lacks significance and scientific interest beyond pointing to a distinct sensitivity/metabolism of mice in comparison to humans given that E4 is already being used as a human contraceptive after clearing all the trials.

Author Response

Point by point answer to reviewers

Reviewer #2

The main concern of the work is the relevance of the results and the biological significance of them. In general terms, and from a clinical point of view, the experimental design goes from mice validation to human test through the very well known and stratified phases of human trials. However, in this work the experimental research flow goes from humans to mice without a rationale thought.   In this regard and based on the data generated through experimental animal models and various clinical studies making use of variable doses and routes of administration, the use of E4 combined with DRPS has already been approved by the European Medicines Agency and the Food and Drug Administration as a new combined oral contraception generation. The data provided in this article lacks significance and scientific interest beyond pointing to a distinct sensitivity/metabolism of mice in comparison to humans given that E4 is already being used as a human contraceptive after clearing all the trials.

Indeed, as mentioned in the introduction of the manuscript and by the reviewer, the use of Estetrol (E4) combined with DRPS has already been approved by the European Medicines Agency and the Food and Drug Administration but for contraception indication only (new combined oral contraception generation).

Nevertheless and very interestingly, the characteristics of E4 makes it an appropriate compound to be used for additional indications such as menopause hormone therapy (MHT) or prevention of hypoxic ischemic encephalopathy in newborns. The use of E4 for these indications will require different formulation or route of administration depending on the specificity of the treatment and its tissue target. Therefore, to properly mimic or anticipate potential new treatments in human, the optimal use of E4 requires a high level of pharmacological knowledge in mice as well. Thus, to reveal new therapeutic applications in humans, to anticipate potential adverse effects and to characterize the complex molecular mechanisms and the pharmacological effects of E4, the accumulation of relevant data from preclinical animal models and translational research is still mandatory.

Since mice share many genetic and physiological similarities with humans, making them an excellent model for studying the effects of drugs or treatments, it is crucial to properly define E4 PK in mice. Especially, patient-derived xenografts (PDX) have emerged as a promising tool allowing the treatment of human tissue samples (cancers and healthy tissues) in in vivo conditions, predicting drug potency. PDX models involve transplanting human tissue directly into immunodeficient mice, submitting human tissue to blood circulating levels of the tested drug. This maintains the original features of patient’s sample and reflects drug sensitivity. Overall, PDX models offer a more clinically relevant approach to preclinical drug testing, which may lead to improved clinical trial design and more effective treatments.

In this context, our study is particularly relevant since it highlights the importance to carefully consider the dose and the route of E4 administration used to design preclinical mouse experiments. The data we are presenting will allow the proper design of preclinical mouse experiments in which it is essential to administer E4 to mice in a pattern that closely mimics or anticipates E4 exposure in women. Therefore, to anticipate potential new treatments or adverse effect in human, the optimal use of hormone therapy requires a high level of pharmacological knowledge in mice as well. This will favor the transposition of results obtained from mice to human.

In addition, the route of administration of a steroid such as Estradiol (E2) largely influences its pharmacokinetic (PK), thus it can be anticipated that E4 will follow this trend. Since mice share many genetic and physiological similarities with humans, making them an excellent model for studying the effects of drugs or treatments, it is crucial to properly define E4 PK in mice. This will allow the proper design of preclinical mouse experiments in which it is essential to administer E4 to mice in a pattern that closely mimics or anticipate E4 exposure in women. For example, this is particularly important to anticipate the harmful effects of menopause treatments on breast cancer. Indeed, the assessment of these treatments on breast cancer risk in women can only be conducted during patient follow-up over decades. Using PDX of breast cancer samples xenografted to mice, it is possible to anticipate with reasonable limitations the potential impact of E4 treatment on human breast cancer if the dose and the route of E4 administration is carefully considered.

Additionally, several preclinical experiments studying the impact of E4 on cardiovascular system, vagina or brain have been conducted using distinct routes of administration such as oral gavage, addition to mouse food, intraperitoneal injection, subcutaneous pellet [1-5]. To properly interpret these preclinical results obtained from mice and to transpose them to human applications, it is mandatory to correctly transpose the dose and the route of administration between mice and women.

Therefore, even if the use of E4 is already approved by EMA and FDA for contraception indication, there is still many preclinical studies to conduct in mice to properly evaluate other indications and to decipher the molecular mechanisms and pharmacological effects of E4. Our results emphasized that the mouse equivalent dose of E4 does not follow the Nair and Jacob allometric correlation. In this context, our study is particularly relevant since it highlights the importance to carefully consider the dose and the route of E4 administration used to design preclinical mouse experiments.

To clarify this issue, we modified the title of the manuscript and we introduced some precision in the introduction. Modifications in the manuscript are highlighted in red.

References

  1. Abot, A., et al., The uterine and vascular actions of estetrol delineate a distinctive profile of estrogen receptor alpha modulation, uncoupling nuclear and membrane activation. EMBO Mol Med, 2014. 6(10): p. 1328-46.
  2. Buscato, M., et al., Estetrol prevents Western diet-induced obesity and atheroma independently of hepatic estrogen receptor alpha. Am J Physiol Endocrinol Metab, 2021. 320(1): p. E19-E29.
  3. Davezac, M., et al., The different natural estrogens promote endothelial healing through distinct cell targets. JCI insight, 2023. 8(5): p.e161284.
  4. Benoit, T., et al., Estetrol, a Fetal Selective Estrogen Receptor Modulator, Acts on the Vagina of Mice through Nuclear Estrogen Receptor alpha Activation. Am J Pathol, 2017. 187(11): p. 2499-2507.
  5. Tskitishvili, E., et al., Estetrol attenuates neonatal hypoxic-ischemic brain injury. Exp Neurol, 2014. 261C: p. 298-307.

Reviewer 3 Report

1. Women and female mice have very different reproduction physiology system, why would you choose female mice to model human E4 exposure? Could you please provide your reasonings?

Author Response

Point by point answer to reviewers

Reviewer #3

Women and female mice have very different reproduction physiology system, why would you choose female mice to model human E4 exposure? Could you please provide your reasonings?

Mice are widely used in reproductive research because they have many similarities with humans in terms of reproductive anatomy, physiology, and genetics. Mice have a short gestational period, which means that researchers can study multiple generations of mice in a relatively short amount of time. They are also relatively easy to manipulate genetically, which allows researchers to create mouse models of human reproductive disorders and diseases. These models can be used to study the underlying mechanisms of these disorders and to test potential treatments. Additionally, the use of mice in research allows for more controlled experiments than would be possible in human studies, reducing variability and increasing the accuracy of results. Overall, the use of mice in reproductive research has led to significant advancements in our understanding of the female reproductive system and has facilitated the development of new treatments and therapies for reproductive disorders and diseases.

Since mice share many genetic and physiological similarities with humans, making them an excellent model for studying the effects of drugs or treatments, it is crucial to properly define E4 PK in mice. Especially, patient-derived xenografts (PDX) have emerged as a promising tool allowing the treatment of human tissue samples (cancers and healthy tissues) in in vivo conditions, predicting drug potency. PDX models involve transplanting human tissue directly into immunodeficient mice, submitting human tissue to blood circulating levels of the tested drug. This maintains the original features of patient’s sample and reflects drug sensitivity. Overall, PDX models offer a more clinically relevant approach to preclinical drug testing, which may lead to improved clinical trial design and more effective treatments.

To reveal new therapeutic applications in humans, to anticipate potential adverse effects and to characterize the complex molecular mechanisms and the pharmacological effects of estetrol, the accumulation of relevant data from preclinical animal models and translational research is still mandatory. In this context, our study is particularly relevant since it highlights the importance to carefully consider the dose and the route of E4 administration used to design preclinical mouse experiments. The data we are presenting will allow the proper design of preclinical mouse experiments in which it is essential to administer E4 to mice in a pattern that closely mimics or anticipates E4 exposure in women. Therefore, to anticipate potential new treatments or adverse effect in human, the optimal use of hormone therapy requires a high level of pharmacological knowledge in mice as well. This will favor the transposition of results obtained from mice to human.

To clarify this issue, we modified the title of the manuscript and we introduced some precision in the introduction. Modifications in the manuscript are highlighted in red.

Round 2

Reviewer 3 Report

I am not convinced by the authors responds.